# Characterization of Biofilm Producing Coagulase-Negative Staphylococci Isolated from Bulk Tank Milk

**DOI:** 10.3390/vetsci9080430

**Published:** 2022-08-13

**Authors:** Yu Jin Lee, Young Ju Lee

**Affiliations:** College of Veterinary Medicine & Zoonoses Research Institute, Kyungpook National University, Daegu 41566, Korea

**Keywords:** staphylococci, coagulase negative staphylococci, milk, biofilm, biofilm associated gene, multidrug resistance

## Abstract

**Simple Summary:**

Coagulase-negative staphylococci are considered less virulent than other variants. However, they have been increasingly recognized as an important cause of bovine mastitis. Moreover, the biofilm-forming ability appears to be important in CoNS pathogenicity, which leads more resistance to antimicrobials. This study investigated the pathogenic potential by assessing the biofilm-forming ability of CoNS isolated from bulk tank milk and analyzed the biogilm-associated resistance to antimicrobial agents. The results indicate that various CoNS isolated from bulk tank milk, not from bovine with mastitis, exhibited a high prevalence of biofilm-forming ability with a high prevalence of MDR, and also biofilm-associated genes with a high prevalence. Therefore, developing a strong monitoring and sanitation program for dairy factories is important to ensure hygienic milk production.

**Abstract:**

Coagulase-negative staphylococci (CoNS) are considered less virulent as they do not produce a large number of toxic enzymes and toxins; however, they have been increasingly recognized as an important cause of bovine mastitis. In particular, the ability to form biofilms appears to be an important factor in CoNS pathogenicity, and it contributes more resistance to antimicrobials. The aim of this study was to investigate the pathogenic potential by assessing the biofilm-forming ability of CoNS isolated from normal bulk tank milk using the biofilm formation assay and to analyze the biofilm-associated resistance to antimicrobial agents using the disc diffusion method. One hundred and twenty-seven (78.4%) among 162 CoNS showed the ability of biofilm formation, and all species showed a significantly high ability of biofilm formation (*p* < 0.05). Although the prevalence of weak biofilm formers (39.1% to 80.0%) was significantly higher than that of other biofilm formers in all species (*p* < 0.05), the prevalence of strong biofilm formers was significantly higher in *Staphylococcus haemolyticus* (36.4%), *Staphylococcus chromogenes* (24.6%), and *Staphylococcus saprophyticus* (21.7%) (*p* < 0.05). Also, 4 (11.4%) among 35 non-biofilm formers did not harbor any biofilm-associated genes, whereas all 54 strong or moderate biofilm formers harbored 1 or more of these genes. The prevalence of MDR was significantly higher in biofilm formers (73.2%) than in non-formers (20.0%) (*p* < 0.05). Moreover, the distribution of MDR in strong or moderate biofilm formers was 81.5%, which was significantly higher than in weak (67.1%) and non-formers (20.0%) (*p* < 0.05). Our results indicated that various CoNS isolated from bulk tank milk, not from bovine with mastitis, have already showed a high ability to form biofilms, while also displaying a high prevalence of MDR.

## 1. Introduction

Staphylococci, a title which includes at least 40 species, are divided into two groups according to their ability to produce the enzyme coagulase: coagulase-negative staphylococci (CoNS) and coagulase-positive staphylococci [1,2,3]. Coagulase-positive staphylococci, including *Staphylococcus aureus*, are a well-known cause of staphylococcal food poisoning, whereas CoNS are considered less virulent as they do not produce a large number of toxic enzymes and toxins compared to coagulase-positive staphylococci [1,2]. However, CoNS have been increasingly recognized as an important cause of bovine mastitis worldwide, with a significant increase in the incidence of intramammary infections in cows based on recent studies [3,4,5]. In particular, the ability to form biofilms appears to be an important factor in CoNS pathogenicity [6]. Biofilm formation occurs when bacteria switch from a planktonic state to a surface-attached state. It ensures bacterial survival by making them less accessible to the host’s defense system [2,7]. Moreover, biofilm exhibit resistance to antimicrobials because these impair their action [8], and act as a chronic source of microbial contamination that may lead to food spoilage in food processing [9]. In Korea, the prevalence and characteristics of CoNS from milk and dairy products have been reported [10,11,12], but there are no reports about their biofilm-forming ability. Thus, the aim of this study was to investigate their pathogenic potential by assessing the biofilm-forming ability of CoNS isolated from normal bulk tank milk, not from bovine with mastitis, and to analyze the biofilm-associated resistance to antimicrobial agents.

## 2. Materials and Methods

### 2.1. CoNS Isolates

A total of 1588 batches of bulk tank milk were collected from 396 dairy farms managed by four companies in Korea. Milk samples were aseptically collected twice each in the summer and winter seasons. Each 50 mL of bulk tank milk was tested for the isolation and identification of *Staphylococcus* spp. according to the standard microbiological protocols published by the Ministry of Food and Drug Safety (2018) [13]. Briefly, 1 mL of each milk sample was cultured in 9 mL of tryptic soy broth with 6% NaCl (BD Biosciences, Sparks, MD, USA). After incubation at 37 °C for 24 h, each medium was streaked onto 5% sheep blood agar (KOMED, Seoul, Korea). Confirmation of *Staphylococcus* spp. was performed using PCR with a species-specific primer as described previously [14]. The classification of CoNS was performed by MALDI-TOF mass spectrometry (Biomerieux, Marcy-l’Étoile, France) based on protein expression profiles using a VITEK MS system (Biomerieux). If two isolates from the same sample origin showed the same antimicrobial susceptibility patterns, only one isolate was randomly chosen. In this study, 162 CoNS were included: 65 *Staphylococcuschromogenes*, 46 *Staphylococcus saprophyticus*, 17 *Staphylococcus xylosus*, 11 *Staphylococcus haemolyticus*, 4 *Staphylococcus simulans*, 5 *Staphylococcus sciuri*, and 14 others.

### 2.2. Biofilm Formation Assay

Biofilm formation was estimated using the standard microtiter plate test, as described with some modifications [15]. In brief, all CoNS isolates were cultured on a brain heart infusion agar (BD Biosciences, Sparks, MD, USA) overnight at 37 °C. Five hundred ul of bacterial suspension adjusted to the 0.5 McFarland standard were inoculated into 3 mL of fresh brain heart infusion broth (BD Biosciences) supplemented with glucose (0.25% wt/vol), and 200 uL of mixture was transferred into 3 wells of a 96 well microtiter plate. After incubation for 18–24 h at 37 °C, planktonic cells were removed by washing with sterile saline. Attached bacteria were fixed with 200 uL of methanol for 15 min, and the bacterial biomass was quantified by measuring the absorbance at 490 nm (A_490_) after staining with safranin solution (0.1% wt/vol) for 10 min and destaining with 50% ethanol−50% glacial acetic acid solution. The ability to form biofilms was classified as negative (A_490_ < 0.110), weak (0.110 ≤ A_490_ < 0.500), moderate (0.500 ≤ A_490_ ≤ 1.500), and strong (A_490_ > 1.500). To certify the analysis, *Staphylococcus epidermidis* ATCC 35984 and *Staphylococcus epidermidis* ATCC 12228 were used as reference strains of strong and weak biofilm producers, respectively, and a sterile medium was used as a contamination control, as described previously [16,17,18].

### 2.3. Antimicrobial Susceptibility Testing

Based on the Clinical and Laboratory Standards Institute guidelines (CLSI, 2018) [19], the antimicrobial resistance of all CoNS isolates was determined using the disc diffusion method with the following discs (BD Biosciences): amikacin (A, 30 µg), ampicillin (AM, 10 µg), amoxicillin-clavulanate (AMC, 20 µg), ceftazidime (CAZ, 30 µg), clindamycin (CC, 2 µg), cefadroxil (CDX, 30 µg), cephalothin (CF, 30 µg), ciprofloxacin (CIP, 5 µg), colistin (CL, 10 µg), cefotaxime (CTX, 30 µg), cefuroxime (CXM, 30 µg), cefazoline (CZ, 30 µg), chloramphenicol (C, 30 µg), doxycycline (DOX, 30 µg), erythromycin (E, 15 µg), cefepime (FEP, 30 µg), cefoxitin (FOX, 30 µg), gentamicin (G, 10 µg), imipenem (IPM, 10 µg), kanamycin (K, 30 µg), oxacillin (OX, 1 µg), penicillin (P, 10 units), tetracycline (TE, 30 µg), teicoplanin (TEC, 30 µg), and vancomycin (VA, 30 µg). *Staphylococcus aureus* ATCC 29213 was used as a quality control. Multidrug resistance (MDR) was defined as an acquired resistance to at least one agent in three or more antimicrobial classes [20].

### 2.4. Detection of Biofilm-Associated Genes

DNA extraction was prepared by the boiling method, as reported [21]. The presence of biofilm-associated genes, such as *aap* (accumulation-associated protein), *atlE* (adhesion and autolysin), *bap* (biofilm-associated protein), *embP* (fibronectin adhesion), *eno* (laminin-binding protein), *fbe* (fibrinogen adhesion), and *icaA* (intercelluar adhesion protein A) was determined by PCR using previously published primer sequences for *aap*, *atlE*, *bap*, *embP*, *eno*, *fbe,* and *icaA* [22,23,24,25].

### 2.5. Statistical Analysis

Statistical analysis using Pearson’s chi-square tests and Fisher’s exact tests with Bonferroni correction was performed in Statistical Package for the Social Science version 25 (SPSS; IBM, Korea). Significant differences were considered at *p* < 0.05.

## 3. Results

### 3.1. Biofilm Formation Potential

The distribution of the ability to form biofilms based on the microtiter plate assay of 162 CoNS isolates is shown in Table 1. One hundred and twenty-seven (78.4%) CoNS showed the ability of biofilm formation, and all species showed a significantly high ability of biofilm formation (*p* < 0.05). Moreover, 73 (45.1%), 23 (14.2%), and 31 (19.1%) among the 127 biofilm-forming isolates were weak, moderate, and strong biofilm formers, respectively. However, the strength to form biofilms showed significant differences in CoNS species. Although the prevalence of weak biofilm formers (39.1% to 80.0%) was significantly higher than that of other biofilm formers in all species (*p* < 0.05), the prevalence of strong biofilm formers was significantly higher in *Staphylococcus haemolyticus* (36.4%), *Staphylococcus chromogenes* (24.6%), and *Staphylococcus saprophyticus* (21.7%), whereas that of moderate biofilm formers was significantly higher in *Staphylococcus chromogenes* (27.7%) and Staphylococcus simulans (25.0%) (*p* < 0.05). The distribution of strength to form biofilms by CoNS species is shown in Figure 1. *Staphylococcus chromogenes* had the highest median value followed by *Staphylococcus haemolyticus*. Although *Staphylococcus saprophyticus* showed a significantly high ability of strong biofilm formation (Table 1), Staphylococcus sciuri had the lowest median value followed by *Staphylococcus saprophyticus*. Moreover, *Staphylococcus haemolyticus* showed the widest deviation range of biofilm formation ability followed by *Staphylococcus chromogenes*, whereas *Staphylococcus xylosus* showed the narrowest deviation range.

### 3.2. Distribution of Biofilm-Associated Genes

The distribution of biofilm-associated genes in 162 CoNS isolates is also shown in Table 1. One hundred and fifty-two (93.8%) isolates harbored at least one of the seven biofilm-associated genes. Among seven biofilm-associated genes, the *eno* gene showed the highest prevalence (50.0% to 82.6%) in all species, except in *Staphylococcus chromogenes* (*p* < 0.05). However, *Staphylococcus chromogenes* carried the *icaA* gene which had the highest prevalence (61.5%) (*p* < 0.05). Moreover, 76 (46.9%) and 64 (39.5%) among 162 isolates carried the *fbe* and *icaA* genes, respectively. In particular, the *fbe* gene appeared in a significantly higher frequency in *Staphylococcus chromogenes*, *Staphylococcus saprophyticus*, and *Staphylococcus sciuri*, whereas the *icaA* gene appeared in a significantly higher frequency in *Staphylococcus chromogenes* and *Staphylococcus saprophyticus* (*p* < 0.05). Forty-nine (30.2%) and thirty (18.5%) isolates carried the *aap* and *atlE* genes, respectively, but their prevalence showed no significant differences among the CoNS species.

### 3.3. Relationship between Biofilm-Associated Genes and Biofilm-Forming Ability

The distribution of the biofilm-associated genes according to the ability of biofilm formation in 162 CoNS isolates is shown in Table 2. A total of 4 (11.4%) among 35 non-biofilm formers did not harbor any of the biofilm-associated genes, whereas all 54 strong or moderate biofilm formers harbored one or more of these genes. In particular, the prevalence of four genes (*aap*, *atlE*, *bap*, and *icaA*) was significantly higher in strong or moderate biofilm formers than in weak and non-formers (*p* < 0.05).

### 3.4. Relationship between MDR and Biofilm-Forming Ability

The distribution of MDR according to the ability of biofilm formation in 162 CoNS isolates is shown in Table 3. The prevalence of MDR was significantly higher in biofilm formers (73.2%) than in non-formers (20.0%) (*p* < 0.05). Moreover, the distribution of MDR in strong or moderate biofilm formers was 81.5%, which was significantly higher than that in weak (67.1%) and non-formers (20.0%) (*p* < 0.05).

## 4. Discussion

Bovine mastitis is the most important disease that leads to economic loss in dairy cattle worldwide [26]. Recently, CoNS are also described as the most common bovine mastitis isolates in many countries, and these emerged as pathogens associated with clinical and subclinical intramammary infection [3,27,28]. Park et al. (2011) [29] reported that *S. chromogenes* (72.2%) was the most distributed CoNS isolate from bovine mastitis in the United States, followed by *Staphylococcus xylosus* (9.1%) and *Staphylococcus haemolyticus* (6.1%). Walid et al. (2021) [30] also reported that most CoNS isolates from bovine mastitis in Egypt were *Staphylococcus epidermidis* (48.4%), *Staphylococcus saprophyticus* (32.3%), and *Staphylococcus haemolyticus* (19.4%). In particular, several CoNS, such as *Staphylococcus epidemidis, Staphylococcus chromogenes*, and *Staphylococcus xylosus*, showed a higher pathogenicity by forming biofilms for bacterial aggregation for a better growth and resistance to adverse conditions [31]. In this study, 162 CoNS isolates, including *Staphylococcus chromogenes* (65 isolates), *Staphylococcus saprophyticus* (46 isolates), *Staphylococcus xylosus* (17 isolates), *Staphylococcus haemolyticus* (11 isolates), *Staphylococcus sciuri* (5 isolates), *Staphylococcus simulans* (4 isolates), and others (14 isolates) were isolated from bulk tank milk, not from bovine with mastitis; however, 127 (78.4%) CoNS isolates showed various abilities to form biofilms. Moreover, 54 (33.3%) CoNS isolates were classified as strong or moderate biofilm formers. Tremblay et al. (2013) and Srednik et al. (2017) [32,33] reported that 48.6% and 44.0% of biofilm-forming CoNS isolates from bovine mastitis in Canada and Argentina, respectively, were classified as strong or moderate biofilm formers. If milk samples were derived from bovine with clinical mastitis rather than from normal bulk tank, a higher prevalence of biofilm formers in CoNS might be confirmed. The prevalence of strong or moderate biofilm formers in this study was significantly higher in *Staphylococcus haemolyticus* and *Staphylococcus chromogenes*, which was similar to previous reports [6,32,33]. The highest median value and widest deviation range were also observed in these two CoNS species. The presence of biofilm-associated genes confers a greater ability to form biofilms [34,35]. In this study, 152 (93.8%) among 162 CoNS isolates harbored at least one or more of the seven biofilm-associated genes. Although 62.3% of 162 CoNS isolates carried the *eno* gene, which showed significantly the highest prevalence, it appears to be distributed regardless of species and biofilm-forming ability, as previous described [2,36]. The *icaA* and *bap* genes are commonly involved in biofilm formation, and their prevalence in this study was 39.5% and 26.5%, respectively. In particular, *Staphylococcus chromogenes* and *Staphylococcus saprophyticus* significantly had a higher prevalence in *icaA*, while *Staphylococcus saprophyticus* significantly had a higher prevalence in *bap*. Interestingly, *Staphylococcus chromogenes* and *Staphylococcus saprophyticus* had a significantly higher prevalence in strong biofilm formers. *Staphylococcus haemolyticus* also had the highest prevalence in strong biofilm formers in this study. However, the prevalence of *Staphylococcus haemolyticus* carrying the *bap* gene was significantly lower than that of *Staphylococcus saprophyticus*, and no *Staphylococcus haemolyticus* isolates harbored the *icaA* gene. Moreover, the prevalence of the *fbe* and *embP* genes, which are involved in surface-adhesins for biofilm formation, was 46.9% and 18.5%, respectively. The highest prevalence of the *fbe* and *embP* genes was observed in *Staphylococcus sciui* and *Staphylococcus simulans*, respectively, for which none of the isolates had a strong biofilm former. The prevalence of four biofilm-associated genes (*aap*, *atlE*, *bap*, and *icaA*) was significantly higher in strong or moderate biofilm formers than weak or non-formers in this study, and other studies have reported a high prevalence of biofilm-associated genes in biofilm-producing staphylococci [37,38,39]. However, the link between the presence of biofilm-associated genes and the ability to form biofilms is not clear and needs to be better understood.

Moreover, the ability to form biofilms is associated with the capacity of bacteria to adhere to a surface and form a layer, so the density of the layer was directly related to the strength of the biofilm produced [40]. Therefore, the strength of biofilm formation was higher in antimicrobial-resistant strains than in antimicrobial-sensitive strains, and a remarkable correlation was found between antimicrobial resistance and biofilm formation strength [41,42,43]. In this study, the prevalence of MDR was significantly higher in biofilm formers than in non-formers. Interestingly, strong and moderate biofilm formers also had a significantly higher prevalence of MDR than weak biofilm formers. Recently, Phophi et al. (2019) [44] reported that biofilm-forming CoNS from mastitis in South Africa showed the significantly higher prevalence in MDR. Moreover, Oliveira et al. (2016) [45] reported that bacteria living in biofilms are up to 1000 times more resistant compared to planktonic bacteria. Therefore, our results support that the biofilm-forming ability limits the treatment strategies for mastitis and might increase morbidity and mortality if biofilm-forming CoNS isolated from bulk tank milk develop as a cause of mastitis. In this study, various CoNS isolated from bulk tank milk, not bovine with mastitis, have already showed their high ability to form biofilms, with a high prevalence of MDR. Therefore, an improved hygiene program should be proposed to control the intramammary infection of environmental bacteria like CoNS.

## Figures and Tables

**Figure 1 vetsci-09-00430-f001:**
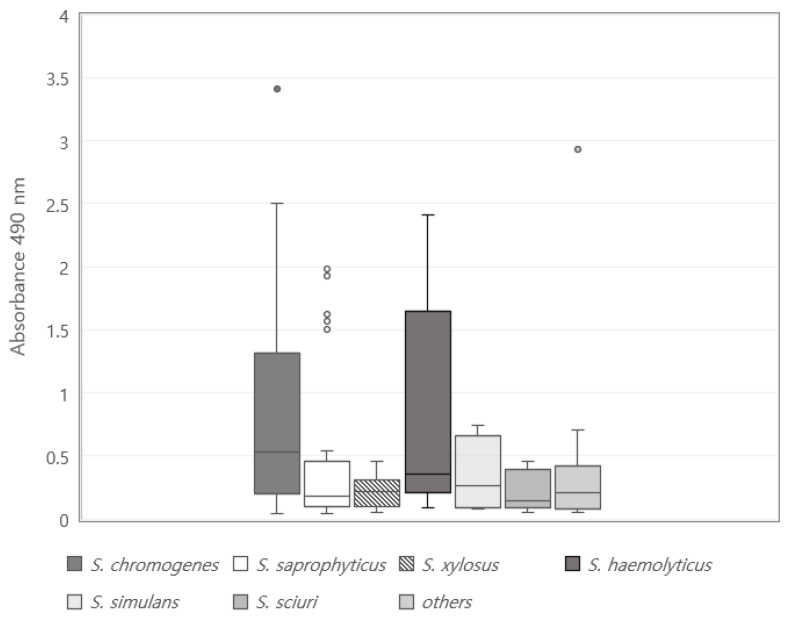
Distribution of strength to form biofilm in 162 coagulase negative staphylococci isolated from bulk tank milk. Points outside the box and whiskers are considered as outliers.

**Table 1 vetsci-09-00430-t001:** Distribution of biofilm formation potential and biofilm-associated genes in 162 coagulase negative staphylococci from milk.

	*Staphylococcus chromogenes* (*n* = 65)	*Staphylococcus saprophyticus* (*n* = 46)	*Staphylococcus xylosus* (*n* = 17)	*Staphylococcus haemolyticus* (*n* = 11)	*Staphylococcus simulans* (*n* = 4)	*Staphylococcus sciuri* (*n* = 5)	Others ^2^ (*n* = 14)	Total (%)
**Biofilm formation** **(A_490_) ^1^**								
Negative	5 (7.7) _c_	17 (37.0) _a_	5 (29.4) _a,b_	1 (9.1) _b,c_	1 (25.0) _a,b_	1 (20.0) _a,b_	5 (35.7) _a_	35 (21.6)
Positive	60 (92.3) _a_*	29 (63.0) _c_*	12 (70.6) _c_*	10 (90.9) _a,b_*	3 (75.0) _b,c_*	4 (80.0) _a,b,c_*	9 (64.3) _c_*	127 (78.4) *
Weak	26 (40.0) _c_^A^	18 (39.1) _c_^A^	12 (70.6) _a,b_^A^	5 (45.5) _c_^A^	2 (50.0) _b,c_^A^	4 (80.0) _a_^A^	6 (42.9) _c_ ^A^	73 (45.1) ^A^
Moderate	18 (27.7) _a_^B^	1 (2.2) _b,c_^C^	0 (0.0) _c_^B^	1 (9.1) _b,c_^C^	1 (25.0) _a_^B^	0 (0.0) _c_^B^	2 (14.3) _a,b_^B^	23 (14.2) ^B^
Strong	16 (24.6) _a,b_^B^	10 (21.7) _a,b_^B^	0 (0.0) _c_^B^	4 (36.4) _a_^B^	0 (0.0) _c_^C^	0 (0.0) _c_^B^	1 (7.1) _b,c_^C^	31 (19.1) ^B^
**Biofilm-associated gene**								
None	4 (6.2) _b,c_^D^	2 (4.3) _b,c_^D^	2 (11.8) _a,b_^C^	1 (9.1) _a,b_^C^	1 (25.0) _a_^B^	0 (0.0) _c_^C^	0 (0.0) _c_^D^	10 (6.2) ^D^
*aap*	20 (30.8) ^B,C^	15 (32.6) ^B,C^	3 (17.6) ^B,C^	4 (36.4) ^B,C^	1 (25.0) ^B^	1 (20.0) ^B^	5 (35.7) ^B,C^	49 (30.2) ^B,C^
*atlE*	12 (18.5) ^C,D^	6 (13.0) ^C,D^	4 (23.5) ^B,C^	2 (18.2) ^C^	1 (25.0) ^B^	1 (20.0) ^B^	4 (28.6) ^C^	30 (18.5) ^C^
*bap*	15 (23.1) _b,c_^C,D^	17 (37.0) _a_^B,C^	2 (11.8) _c_^C^	3 (27.3) _b,c_^B,C^	1 (25.0) _b,c_^B^	0 (0.0) _c_^C^	5 (35.7) _a,b_^B,C^	43 (26.5) ^C^
*embP*	14 (21.5) _b,c_^C,D^	5 (10.9) _b,c_^C,D^	3 (17.6) _b,c_^B,C^	1 (9.1) _c_^C^	1 (25.0) _a,b_^B^	0 (0.0) _c_^C^	6 (42.9) _a_^B,C^	30 (18.5) ^C^
*eno*	27 (41.5) _c_^B,C^	38 (82.6) _a_^A^	11 (64.7) _a,b_^A^	9 (81.8) _a_^A^	2 (50.0) _b,c_^A^	4 (80.0) _a_^A^	10 (71.4) _a,b_^A^	101 (62.3) ^A^
*fbe*	32 (49.2) _a,b_^B,C^	25 (54.3) _a,b_^B^	7 (41.2) _b,c_^B^	3 (27.3) _c_^B,C^	1 (25.0) _c_^B^	3 (60.0) _a_^B^	5 (35.7) _b,c_^B,C^	76 (46.9) ^B^
*icaA*	40 (61.5) _a_^A^	15 (32.6) _a,b_^B,C^	4 (23.5) _b,c_^B,C^	0 (0.0) _c_^D^	1 (25.0) _b,c_^B^	1 (20.0) _b,c_^B^	3 (21.4) _b,c_^C^	64 (39.5) ^B,C^

The superscript letter represents significant difference of the column, while the subscript letter represents significant difference of the row (*p* < 0.05). * Statistically significant difference between biofilm-positive isolates and biofilm-negative isolates (*p* < 0.05). ^1^ A_490_ = Absorbance at 490 nm. ^2^ Others: Staphylococcus arlettae (*n* = 1), Staphylococcus capitis (*n* = 3), Staphylococcus cohnii (*n* = 2), Staphylococcus epidermidis (*n* = 1), Staphylococcus equorum (*n* = 2), Staphylococcus gallinarum (*n* = 2), Staphylococcus lentus (*n* = 1), Staphylococcus succinus (*n* = 2).

**Table 2 vetsci-09-00430-t002:** Relationship between biofilm-associated genes and biofilm-forming ability in 162 coagulase negative staphylococci from milk.

Antimicrobial Resistance	Biofilm Producer		
	Strong or Moderate Biofilm Former (*n* = 54)	Weak Biofilm Former (*n* = 73)	Non-Former (*n* = 35)
Non-MDR	10 (18.5) _c_^B^	24 (32.9) _b_^B^	28 (80.0) _a_^A^
MDR	44 (81.5) _a_^A^	49 (67.1) _b_^A^	7 (20.0) _c_^B^

The superscript letter represents significant difference of the column, while the subscript letter represents significant difference of the row (*p* < 0.05).

**Table 3 vetsci-09-00430-t003:** Relationship between multidrug resistance (MDR) and biofilm-forming ability in 162 coagulase negative staphylococci from milk.

Biofilm-Associated Gene	Biofilm Producer		
	Strong or Moderate Biofilm Former (*n* = 54)	Weak Biofilm Former (*n* = 73)	Non-Former (*n* = 35)
None	0 (0.0) _b_^E^	6 (8.2) _a,b_^D^	4 (11.4) _a_^C,D^
*aap*	28 (51.9) _a_^A,B,C^	16 (21.9) _b_^B,C^	5 (14.3) _c_^C,D^
*atlE*	18 (33.3) _a_^C,D^	9 (12.3) _b_^C,D^	3 (8.6) _c_^D^
*bap*	22 (40.7) _a_^B,C,D^	16 (21.9) _b_^B,C^	5 (14.3) _c_^C,D^
*embP*	12 (22.2) _a,b_^D^	9 (12.3) _b_^C,D^	9 (25.7) _a_^B,C^
*eno*	35 (64.8) _a,b_^A^	38 (52.1) _b_^A^	28 (80.0) _a_^A^
*fbe*	32 (59.3) _a_^A,B^	28 (38.4) _b_^A,B^	16 (45.7) _a,b_^B^
*icaA*	35 (64.8) _a_^A^	24 (32.9) _b_^A,B^	5 (14.3) _c_^C,D^

The superscript letter represents significant difference of the column, while the subscript letter represents significant difference of the row (*p* < 0.05).

## Data Availability

Not applicable.

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
