# Peer review of "Characterization of Biofilm Producing Coagulase-Negative Staphylococci Isolated from Bulk Tank Milk"

_vetsci, 2022, doi:10.3390/vetsci9080430_

Round 1

Reviewer 1 Report

The authors have addressed the majority of changes that I suggested following the original submission and the revised manuscript is much improved compared to the original submission. The revised manuscript does require a significant number of minor corrections and changes to the text before it can be accepted for publication in Veterinary Sciences, as follows:

Throughout the manuscript, italicise all names of bacterial species

Line 19.  Change “rec-ognized” to “recognized”

Line 21.  Change “antimicro-bials” to “antimicrobials”

Line 22.  Change “bio-film” to “biofilm”

Line 24.  Insert “the” before “disc”

Line 32.  Change “prev-alence” to “prevalence”

Line 35.  Change “in-dicated” to “indicated”

Line 38.  Change “Coagulase” to “coagulase”

Line 62.  Change “ac-cord-ing” to “according”

Line 63.  Change “staphy-lococci” to “staphylococci”

Line 64.  Change “staph-ylococci” to “staphylococci”

Line 67.  Change “staphylo-cocci” to “staphylococci”

Line 72.  Insert “a” before “planktonic” and insert “a” before “surface”

Line 73.  Change “sys-tem” to “system”

Line 77.  Change “re-ports”

Line 93.  Change “sus-ceptibility” to “susceptibility”

Lines 113-114.  Remove “de-scribed”

Line 116.  Change “inoculat-ed” to “inoculated”

Line 121.  Change “ab-sorbance” to “absorbance”

Line 127.  Change “con-tamination” to “contamination” and remove “described”

Line 143.  Change “diffu-sion” to “diffusion”

Line 152.  Change “re-sistance” to “resistance”

Line 154.  Change “Biofilm” to “biofilm”

Line 163.  Insert “the” before “boiling” and remove “described”

Line 164.  Italicise “aap” and “atlE”

Line 165.  Italicise “bap” and “embP” and change “adhe-sion” to “adhesion”

Line 166.  Italicise “eno”, “fbe” and “icaA”

Figure 1.  The numbers on the y-axes need changing to English numbers

Figure 1.  On the bottom graph the label on the last data set is missing

Line 207.  What are “whiskers”?, change “outlier” to “as outliers”

Line 212.  Change “bio-film” to “biofilm” and italicise “eno”

Line 214.  Italicise “icaA”

Line 216.  Italicise “fbe”, “icaA” and “fbe”

Line 217.  Italicise “icaA”

Line 219.  Italicise “aap” and “atlE”

Line 220.  Change “respec-tively” to “respectively”

Line 224.  Change “infor-mation” to “infor-mation”

Line 227.  Change “prev-alence” to “prevalence” and italicise “aap”, “atlE”, “bap” and “icaA”

Line 246.  Change “iso-lates” to “isolates”

Line 324.  Change “masti-tis” to “mastitis”

Line 331.  Change “re-sistance” to “resistance”

Line 332.  Change “chro-mogenes” to “chromogenes”

Line 339.  Change “respec-tively” to “respectively”

Line 340.  Change “de-rived” to “derived”

Line 345.  Change “pres-ence” to “presence”

Line 348.  Italicise “eno”

Line 350.  Change “previous” to “previously” and italicise “icaA” and “bap”

Line 352.  Change “sap-rophyticus” to “saprophyticus” and italicise “icaA”

Line 353.  Change “signifi-cantly” to “significantly” and italicise “bap”

Line 356.  Change “preva-lence” to “prevalence” and italicise “bap”

Line 358.  Italicise “icaA”, “fbe” and “embP”

Line 359.  Change “bio-film” to “biofilm”

Line 360.  Italicise “fbe” and “embP”

Line 362.  Italicise “aap”, “atlE”, “bap” and “icaA”

Line 366.  Change “bio-films” to “biofilms”

Line 368.  Change “ad-here” to “adhere”

Line 371.  Change “for-mation” to “formation”

Line 380.  Change “de-velop” to “develop”

Line 383.  Change “pro-posed” to “proposed”

Line 389.  Change “acqui-sition” to “acquisition”

Author Response

All typos, missing words, and italics have been corrected.

Figure 1.  The numbers on the y-axes need changing to English numbers

- If it is written in English numbers, the figure 1 does not look simple.

Figure 1.  On the bottom graph the label on the last data set is missing

-I'm sorry that I kept editing this, but I don't know why the content disappears once I submit it. Once again, I edited it and sent it back.

Reviewer 2 Report

The authors still did not in any way respond to the previous suggestion and did not complete the information regarding the animals from which the samples were taken. So it's still a large collection of completely random trials. Similarly, for the data on drug susceptibility - there are still no data available. As the authors did not consider my suggestions, my opinion on this manscript is still negative and I do not recommend its publication for publication in its current form.

 No italics in the names of the Staphylococcus species

 Please verify the language in Fig 1

Author Response

Thank you for your comment.

I edited the part you asked me to correct, but the manager said there was an error in the system and the revisions may not have been put in correctly. I sent you again the revised part with attachments. The part about italics was also modified, and the part of figure1 was also modified.

However, I'm sorry that I'm not sure exactly what reviewers are trying to say about drug susceptibility.

And, you also said that the sample size is randomly large. In my opinion of the relatively large sample size, the random selection experiment is a little more fair and transparent because the sample is so many and varied. Similarly, in a presidential election, people are randomly asked for their voting results, and the results of an exit poll are announced, and the results are identical to the actual presidential election results. Could you please explain why a random sample is a problem?

Reviewer 3 Report

Dear authors,

The present manuscript is well written but is lacking of global interest and presents expected results. My only suggestion is that the sample size need justification and eligible criteria of the participants

Further more even if the sample size is relative big it should be discussed if the period of sampling or other potential factors may had an affect on microbial population.

Author Response

I don't know if I answered the reviewer's comment properly, but if I said something different, please say it again. Thank you for your comment.

Matters related to the sampling period have been added to the manuscript, and the typo you pointed out has been corrected.

And in my opinion about the relatively large sample size, I think that random selection experiments are a little more fair and transparent because there are many and varied samples. As a similar example, in the presidential election, people are randomly asked about the results of the vote and the results of the exit poll are announced.

Round 2

Reviewer 2 Report

The authors only referred to technical comments, therefore it is difficult for me to accept the manuscript. It follows that the authors tested random samples of milk (and it is important whether the cows had clinical, subclinical symptoms or were treated before sampling), correlating them with the ability to produce biofilm and multi-drug resistance, which is influenced by the factors I mentioned. They also did not present drug resistance profiles, although they cite that the study was conducted - demonstrating only MDR – it is confusing and does not add anything in my opinion.

With all due respect, the explanation of the authors and comparing the sampling to the presidential election has nothing to do with it, and in this situation I cannot recommend the manuscript for publication. 

Sorry for the disappointment, but in this situation I cannot change my opinion without introducing significant changes by the authors

Author Response

First, thank you very much for your honest comments of my report, and I apologize in advance for possible misunderstandings due to my poor English.

I gave an example related to the presidential election in the sense that broad samples lead to more objective results to the question of why you are doing a wide sample. I'm sorry if I misunderstood the meaning of the question.

There is still parts I don't understand, so could I ask the reviewer a question? Regarding the question you have given in detail this time(random samples of milk), in the middle of the "introduction", I explained that the CoNS isolates are recognized as an important cause of mastitis in bovine in many recent studies with references, and the incidence of intramammary infection is increasing. In addition, at the end of the "introduction", it is said that normal bulk tank milk, not from bovine with mastitis, milk was obtained from bovine without prior mastitis symptoms, and therefore it was mentioned that the milk used in this report is not related to mastitis-related antimicrobials. I think I explained enough about what kind of milk sample it was in "introduction" in my report. So, Could I ask if there is anything else I need to improve on here?

As for MDR, as reviewers know better, if bacteria are resistant to three or more antibiotics in antimicrobial disc test, it is that bacteria have MDRs. The definition of MDR and the list of antimicrobials for the antimicrobials disc test I used are in "Materials and methods 2.3". The reason why I related each antimicrobials to MDR rather than biofilm is given in the "last paragraph of the Discussion". The bacteria forms a biofilm, and a biofilm layer is formed on the surface of the bacteria. Then, the antimicrobials will not penetrate, and the bacteria will protect themselves. The stronger the biofilm, the stronger the layer, the more the antibiotic cannot work. At first, it is resistant to only one antibiotic, then it becomes resistant to more and more antibiotics, and the bacteria develops MDR. In this report and the reports included as references, the main point is that biofilm-forming bacteria have a strong MDR, and the higher the MDR, the weaker the effect of antibiotics, so it is difficult to treat mastitis in bovine. Although resistance to each antibiotic is not serious, it is difficult to treat bacteria that form biofim with MDR with antibiotics, so, in this report, I wanted to talk about MDR and bacteria with biofilms. Also, if I have a chance, I will go into more detail about what the reviewer said in the next report. If my explanation is insufficient, please feel free to ask again. Thank you.

This manuscript is a resubmission of an earlier submission. The following is a list of the peer review reports and author responses from that submission.

Round 1

Reviewer 1 Report

General comment

Manuscript Animals-1733003 (“Characterization of Biofilm Producing Coagulase-Negative Staphylococci Isolated From Bulk Tank Milk”) describes and discusses the findings on the pathogenic potential of CoNS isolates from normal milk by assessing the biofilm-forming ability and the biofilm-associated resistance to antimicrobials.

The topic is very interesting, the applied methodology is generally consistent, and the resulting conclusions contribute to this scientific field. Only one issue needs to be clarified in the methodology; the procedure by which the isolates were selected for further examination. Was the antimicrobial susceptibility testing carried out first? How many isolates were tested? (See specific comments).

The manuscript is well written, is well-structured and adequately cited. I have the bellow comments for the authors’ consideration.

Materials and Methods

Lines 56-57: Ιt would be interesting the prevalence of CoNS in bulk tank milks samples and/or farms to be presented and the relative results would be of interest in the discussion.

Lines 61-63: “If two isolates from the same origin showed the same antimicrobial susceptibility patterns, only one isolate was randomly chosen”.

Τhe procedure by which the isolates were selected for further examination is not clear. Was the Antimicrobial susceptibility testing carried out first? Ηow many isolates were tested?

Should be considered that isolates with the same antimicrobial susceptibility patterns may harbor different biofilm-associated genes. Consider this issue in the discussion.

Lines 63-64. In this study, 162 CoNS were included: 65 S. chromogenes, 46 S. saprophyticus, 17 S. xylosus, 11 S. haemolyticus, 4 S. simulans, 5 S. sciuri, and 14 others. Were these strains isolated or further tested? see above comment.

Table 1.  It is suggested that Table 1 could be deleted since all primers are published ones. The text could be modified accordingly e.g.…. “using previously published primer sequences for aap, atlE, embP, fbe and icaA (Rohde et al., 2005)….”

Tables 2 and 3. See the titles of the tables in relation to the results presented and the order in which they are presented in the text. In Table 3 the results of the Table 4 are presented and vice versa.

Table 2,3,4 & Figure 1. “CoNS” “instead of “coagulase negative staphylococci”.

Lines 134-135. “S.” instead of “Staphylococcus”.

Reviewer 2 Report

I regret to inform that I do not recommend the manuscript in its current form for publication.

First of all, the manuscript, despite the fact that it has been conducted on a fairly large group of samples, does not provide any innovative information.

The research group was not defined; it is known that the milk samples came from animals without symptoms of mastitis from 396 farms (the method of sampling, the clinical status of the animals, age and criteria related to the use of previous antibiotics are unknown). Therefore, it is difficult to infer any regularities resulting from the obtained results. The purpose of these studies is also unclear (to assess the biofilm production capacity of randomly isolated strains?)

The authors write about the correlation with drug resistance of the studied strains, however, the drug resistance profiles obtained by them were not even reported. The results only include the division into MDR and non-MDR strains.

Abstract: line 20: what does "normal milk" mean?

Keywords: there is no need to use the words Staphylococci and coagulase negative staphylococci twice, similarly, biofilm and biofilm associated gene

Introduction: Please update information on the number of Staphylococcus species. The current form of the description shows that only S aureus belongs to the coagulase positive species - a redrafting is required

Materials and methods: The Ministry of Food and Drug Safety protocols are not widely available, so the process of isolation and identification of strains should be described in more detail. Especially that coagulase-negative species of Staphylococcus in particular are quite problematic to identify. It is also unclear whether the statement in line 62 is for two isolates belonging to the same species or if the authors typed only one strain from each sample or more?

Line 79 and others: please verify the accuracy of spelling the reference strains

Results: no description of the susceptibility results. Table 4 is not clear. Table 3 shows that 100 strains that met the MDR criteria were selected. In table 4, although it only applies to MDR strains, there are a total of 162 strains (all)

Most of the discussion is a repetition of the results - it definitely requires editing. Line 222-227: The sentence is not understood and the authors did not explain how MDR is related to the ability to produce biofilm.

The conclusion is imprecise and boils down to the necessity to control the presence of bacteria in animals

Reviewer 3 Report

In this manuscript the authors have investigated the biofilm-forming ability of coagulase-negative staphylococci (CoNS) isolated from 1,588 samples of bulk tank milk from 396 dairy farms in Korea and analyzed their biofilm-associated resistance to antimicrobials. CoNS have been recognized as an important cause of bovine mastitis. Out of 162 CoNS isolates, 127 (78.4%) showed various abilities to form biofilms with 54 (33.3%) classified as strong or moderate biofilm formers. At least one of the seven biofilm-associated genes were found in 152 (93.8%) out of the 162 CoNS isolates. It was also observed that MDR was significantly higher in biofilm formers than in non-formers, and strong and moderate biofilm formers had a significantly higher prevalence of MDR than weak biofilm formers.

I suggest that this manuscript could be suitable for publication in Animals if the text is subjected to a substantial number of minor changes, as follows:

Lines 20 and 23.  Avoid beginning a sentence with numerals

Line 30.  Change “Staphylococci” to “staphylococci” and “Coagulase” to “coagulase”

Line 44.  Insert “a” before “planktonic” and insert “a” before “surface-attached”

Line 45.  Change “biofilm” to “biofilms”

Line 50.  Remove “of”

Line 68.  Change “reported” to “previously”

Line 81.  Change “reported” to “previously”

Line 97.  Change “prepared by boiling” to “performed by the boiling” and “reported” to “previously”

Line 103.  Change “Primers” to “PCR primers”

Line 108.  Change “at” to “as”

Lines 131-132.  Change “The superscript letter represents significant difference of the column, while the subscript letter represents significant difference of the row” to “The superscript letters represent significant differences in a column, while the subscript letters represent significant differences in a row”

Line 139.  Change “of strength to form biofilm” to “of the strength to form biofilms”

Line 140.  What are “whiskers”?  Change “outlier” to “outliers”

Line 162.  Change “from milk” to “from bulk tank milk”

Lines 164-165.  Change “The superscript letter represents significant difference of the column, while the subscript letter represents significant difference of the row” to “The superscript letters represent significant differences in a column, while the subscript letters represent significant differences in a row”

Line 173.  Change “from milk” to “from bulk tank milk”

Lines 174-175.  Change “The superscript letter represents significant difference of the column, while the subscript letter represents significant difference of the row” to “The superscript letters represent significant differences in a column, while the subscript letters represent significant differences in a row”

Line 180.  Change “et al., (2011)” to “et al. (2011)”

Line 182.  Change “et al (2021)” to “et al. (2021)”

Line 192.  Change “et al., (2013)” to “et al. (2013)” and “et al., (2017)” to “et al. (2017)”

Line 202.  Avoid beginning a sentence with numerals

Line 204.  Change “previous” to “previously”

Line 230.  Change “et al., (2019)” to “et al. (2019)”

Line 231.  Change “the” to “a”

Line 232.  Change “et al., (2016)” to “et al. (2016)”

Line 257.  Italicise “Staphylococcus epidermidis”

Line 259.  Italicise “Staphylococcus aureus”

Lines 259-260.  Italicise “Staphylococcus argenteus”

Line 260.  Italicise “Staphylococcus schweitzeri”

Line 274.  Italicise “Staphylococcus aureus”

Line 292.  Change “Bulk” to “bulk”

Line 296.  Italicise “Staphylococcus aureus”

Lines 297-298.  Italicise “Pseudomonas aeruginosa”

Line 300.  Italicise “Staphylococcus epidermidis”

Line 304.  Italicise “Staphylococcus aureus” and “Staphylococcus epidermidis”

Line 307.  Italicise “Staphylococcus aureus”

Line 315.  Italicise “Mycobacterium bovis”

Lines 315-116.  Italicise “Mycobacterium tuberculosis”

Line 319.  Italicise “Staphylococcus aureus”

Line 322.  Italicise “Staphylococcus epidermidis”

Lines 325-326.  Italicise “Staphylococcus epidermidis”

Line 326.  Italicise “Staphylococcus aureus”

Line 366.  Italicise “Staphylococcus aureus”

Line 373.  Italicise “Klebsiella pneumoniae”

Line 366.  Change “staphylococcus” to “Staphylococcus” and italicise “Staphylococcus aureus”